# Generating single-cell gene expression profiles for high-resolution spatial transcriptomics based on cell boundary images

Bohan Zhang[1,2,†], Mei Li[1,3,†], Qiang Kang[1,†], Zhonghan Deng[1], Hua Qin[2], Kui Su[1], Xiuwen Feng[1], Lichuan Chen[1], Huanlin Liu[1], Shuangsang Fang[2], Yong Zhang[1], Yuxiang Li[1], Susanne Brix[3,*] and Xun Xu[1,*]

1 BGI Research, Shenzhen, 518083, China
2 BGI Research, Beijing, 102601, China
3 Department of Biotechnology and Biomedicine, Technical University of Denmark, 2800, Kgs. Lyngby, Denmark

**Submitted:** 27 December 2023

\* Corresponding authors. E-mail: sbrix@dtu.dk; xuxun@genomics.cn

† Contributed equally.

Preprint submitted at https://doi.org/10.1101/2023.12.25.573324

Included in the series: *Spatial Omics: Methods and Application* (https://doi.org/10.46471/GIGABYTE_SERIES_0005)

## ABSTRACT

In spatially resolved transcriptomics, Stereo-seq facilitates the analysis of large tissues at the single-cell level, offering subcellular resolution and centimeter-level field-of-view. Our previous work on StereoCell introduced a one-stop software using cell nuclei staining images and statistical methods to generate high-confidence single-cell spatial gene expression profiles for Stereo-seq data. With advancements allowing the acquisition of cell boundary information, such as cell membrane/wall staining images, we updated our software to a new version, STCellbin. Using cell nuclei staining images, STCellbin aligns cell membrane/wall staining images with spatial gene expression maps. Advanced cell segmentation ensures the detection of accurate cell boundaries, leading to more reliable single-cell spatial gene expression profiles. We verified that STCellbin can be applied to mouse liver (cell membranes) and *Arabidopsis* seed (cell walls) datasets, outperforming other methods. The improved capability of capturing single-cell gene expression profiles results in a deeper understanding of the contribution of single-cell phenotypes to tissue biology.

**Availability & Implementation:** The source code of STCellbin is available at https://github.com/STOmics/STCellbin.

**Subjects** Genetics and Genomics, Bioinformatics, Developmental Biology

## STATEMENT OF NEED

Spatially resolved single-cell transcriptomics enables the generation of comprehensive molecular maps that provide insights into the spatial distribution of molecules within individual cells constituting tissues. This groundbreaking technology offers insights into the location and function of cells across diverse tissues, advancing our understanding of organ development [1], tumor heterogeneity [2], cancer evolution [3], and other biological mechanisms. Resolution and field-of-view are critical parameters in spatial transcriptomics. Specifically, a high resolution provides detailed molecular information at the single-cell level, and a large field-of-view facilitates the creation of complete 3D maps, capturing

biological functions at the organ level. Stereo-seq simultaneously achieves subcellular resolution and a centimeter-level field-of-view, providing the technical foundation for obtaining comprehensive spatial gene expression profiles of whole tissues at the single-cell level [4]. Our previous work introduced StereoCell, a one-stop software for obtaining single-cell spatial gene expression profiles with a high signal-to-noise ratio from Stereo-seq data [5]. StereoCell takes the cell nuclei staining image tiles and its corresponding spatial gene expression data as input, and it performs tasks such as image stitching, image registration, tissue segmentation, cell nuclei segmentation, and molecule labeling steps. Notably, Stereo-seq uses cell nuclei staining images; however, there exists a significant difference between cell nuclei and cell boundary staining images, based on cell membrane/wall staining, in terms of the ability to capture robust and precise cell-specific gene expression profiles. Despite the widespread use of spatial techniques, such as MERFISH [6], CosMx [7], and Xenium [8], several of these techniques struggle to provide accurate cell boundary information, as they rely on cell nuclei staining images generated using stains such as 4,6-diamidino-2-phenylindole (or DAPI). Hematoxylin-eosin and single-strand DNA fluorescence nuclei staining images are also commonly used and readily obtainable. The updated Stereo-seq technology incorporates a procedure leveraging simultaneous cell membrane/wall and cell nuclei staining by adding multiplex immunofluorescence (mIF) and calcofluor white (CFW) staining [9, 10], enabling to acquire more accurate cell boundary information automatically and, consequently, more reliable single-cell spatial gene expression profiles.

Here, we updated StereoCell to a new version: STCellbin. The new version retains key steps from StereoCell, such as image stitching, tissue segmentation, and molecule labeling. Additionally, it incorporates improved image registration and cell segmentation steps. Notably, the "track line", a crossed linear marker embedded on the Stereo-seq chip, is key to the image registration step of StereoCell [5]. As the cell membrane/wall staining images miss the "track line" information, the cell nuclei staining images are used to align the cell membrane/wall staining images with the spatial gene expression maps, thereby obtaining registered cell boundary information in the cell segmentation step. Based on the cell boundary information, STCellbin directly assigns the molecules to their corresponding cells, obtaining single-cell spatial gene expression profiles. We applied STCellbin to mouse liver (cell membrane) and *Arabidopsis* seed (cell wall) datasets and confirmed the accuracy of the cell segmentation provided by the software. This update offers a comprehensive workflow to obtain reliable single-cell spatial gene expression profiles based on cell membrane/wall information. Hence, STCellbin provides support and guidance, particularly for scientific investigations based on Stereo-seq data.

## IMPLEMENTATION

### Overview of STCellbin

The process of STCellbin includes image stitching, image registration, cell segmentation, and molecule labeling (Figure 1). Input into STCellbin includes Stereo-seq spatial gene expression data, alongside cell nuclei and cell membrane/wall staining image tiles. The stitched cell nuclei and cell membrane/wall staining images are obtained using the MFWS algorithm [5]. These two stitched staining images are registered using a Fast Fourier Transform (FFT) algorithm [11]. The spatial gene expression data is transformed into a map, which is then registered with a stitched cell nuclei staining image based on "track line"

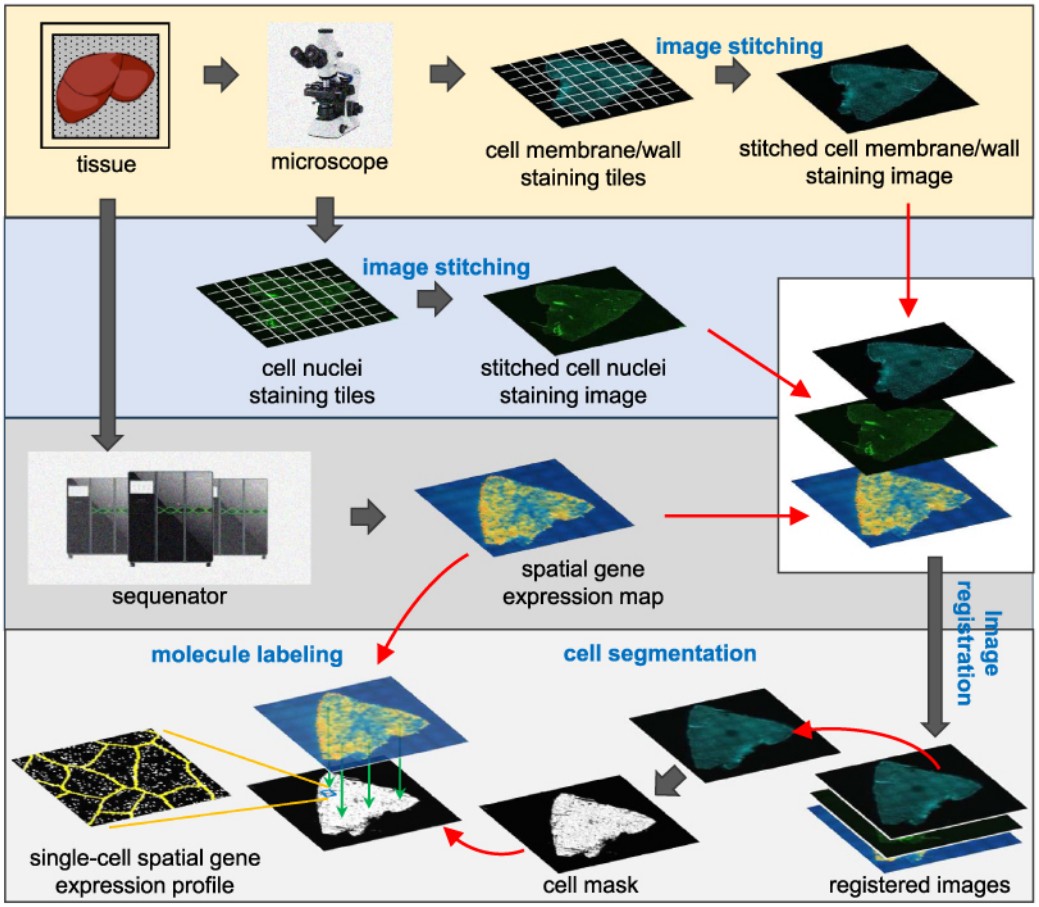

**Figure 1.** Overview of STCellbin. The cell nuclei and cell membrane/wall staining image tiles are stitched into individual large images. The spatial gene expression map and stitched cell membrane/wall staining image are registered using the stitched cell nuclei staining image as a bridge. The cell mask is directly obtained from the registered cell membrane/wall staining image by cell segmentation. The single-cell spatial gene expression profile is obtained by overlaying the generated cell mask and the gene expression map.

information. Thus, the registration of the gene expression map and cell membrane/wall staining image is implemented. Cell segmentation is performed on the registered cell membrane/wall staining image using the adjusted Cellpose 2.0 tool [12] to obtain the cell mask. Molecules are then assigned to their corresponding cells based on the cell mask, thus generating the single-cell spatial gene expression profile. The tissue segmentation step based on Bi-Directional ConvLSTM U-Net [13] is set as optional, and it can be used to generate a tissue mask to assist in filtering out impurities outside the tissue.

## Image stitching

The image stitching step in STCellbin is consistent with the one in StereoCell. The MFWS algorithm [5] leverages FFT [11] to compute offsets between adjacent tiles featuring overlapping areas. This enables the stitching of these tiles, and the process is extended iteratively to encompass all tiles in the dataset. The relative error, absolute error, and computational efficiency of MFWS were assessed in our previous work [5].

## Image registration

The registration of STCellbin includes three stages. The first stage is the registration of the stitched cell nuclei and stitched cell membrane/wall staining images. These two staining images have similar sizes and no significant difference in rotation because the chip does not move when they are photographed. The key to this registration is to calculate their offset. The size of the cell membrane/wall staining image is adjusted to match that of the cell nuclei staining image through cutting and zero-padding (Figure 2A). The two staining images are mean-based subsampled [14] (Figure 2B). The offset of the subsampled images is calculated through FFT [11], similarly to MFWS [5] (Figure 2C). Then, the calculated offset is restored to the scale of the original images (Figure 2D). Thus, these two staining images can be registered.

The second stage is the registration of the stitched nuclei staining image and spatial gene expression map. This registration is the same as in StereoCell [5]. The spatial gene expression data is transformed into a map. The stitched cell nuclei staining image is registered with the map based on "track line" information, involving scaling, rotating, flipping, and translating on the stitched cell nuclei staining image.

The third stage is the registration of the stitched cell membrane/wall staining image and the spatial gene expression map. Since the cell nuclei and cell membrane/wall staining images have been registered in the first stage, the same operations of the second stage, including scaling, rotating, flipping, and translating, are applied to the cell membrane/wall staining image (Figure 2E). Then, the cell membrane/wall staining image and spatial gene expression map can be registered. Moreover, when utilizing staining images produced with a multi-channel microscope, STCellbin can omit the registration among these images. STCellbin can also process the case of multiple mIF staining images captured from identical tissues using the same microscope when there is only a difference in offsets among these images.

## Cell segmentation

The cell segmentation step of STCellbin uses Cellpose 2.0 [12] with some adjustments. The model architecture of Cellpose 2.0 and its weight files "cyto2" are downloaded. However, due to the large size of the staining images derived from Stereo-seq data, Cellpose 2.0 cannot be executed smoothly using normal hardware configurations. To address this issue, the staining images are cropped into multiple tiles with overlapping areas to perform cell segmentation and record the coordinates of these tiles. The overlapping areas prevent cells at the border of the tiles from being cropped. For optimal results, segmentations with different values of the cell diameter are performed independently, and the segmentation yielding the highest total cell area is retained. Next, all the segmented tiles are assembled into the final segmented result according to the recorded coordinates. Moreover, when selecting the tissue segmentation option, an additional step involves applying a filter to the cell mask using the tissue mask, resulting in a refined segmented output.

## Molecule labeling

The molecule labeling of STCellbin is the same as in StereoCell, in principle. StereoCell assigns molecules from the cell nuclei to the cell by using the cell nuclei mask, and then assigns molecules outside the cell nuclei to the cells with the highest probability density using a Gaussian Mixture Model [15]. Conversely, STCellbin directly assigns molecules to the

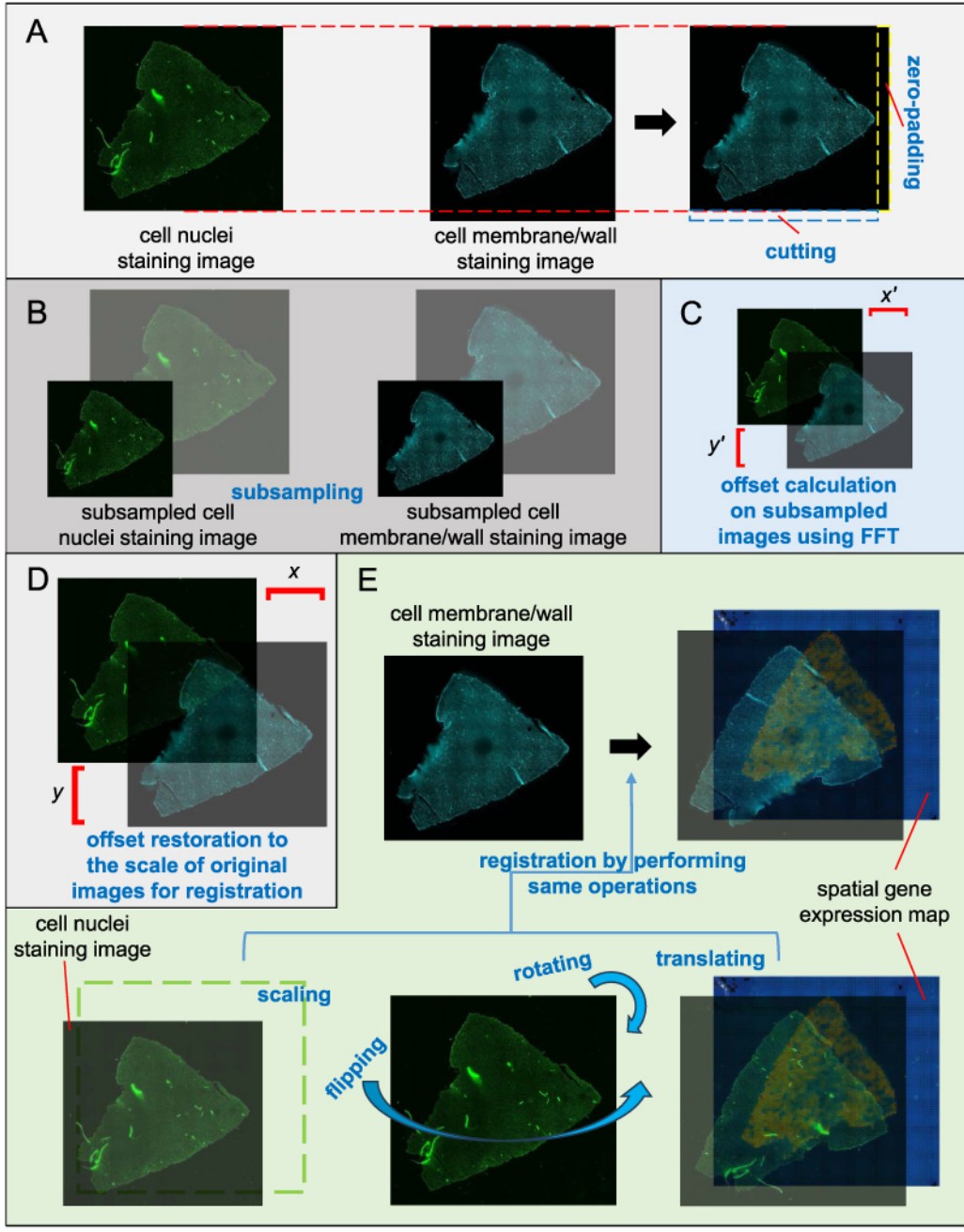

**Figure 2.** Registration of the cell membrane/wall staining image and spatial gene expression map using the cell nuclei staining image as a bridge. (A) The size of the cell membrane/wall staining image is adjusted to be consistent with the cell nuclei staining image. (B) The cell nuclei and cell membrane/wall staining images are subsampled. (C) Calculation of the offsets of the subsampled images. (D) Restoring the offsets to the scale of the original images for registration. (E) Registration of the spatial gene expression map and cell nuclei staining image by performing scaling, rotating, flipping, and translating, followed by the registration of the spatial gene expression map and cell membrane/wall staining image by performing the same operations.

**Table 1.** Details of two datasets used for evaluation of cell boundary information.

| Detail | Mouse liver dataset | *Arabidopsis* seed dataset |
|---|---|---|
| Data source | A slice of liver | Slices of multiple seeds |
| Cell nuclei dye | DAPI | ssDNA |
| Cell membrane/wall dye | mIF | CFW |
| Number of molecules | 16,177,288 | 62,884,637 |

cells based on the cell mask, while assigning molecules outside the cell is optional. This decision was driven by the observation that cell membranes/walls are usually tightly packed, with a few molecules appearing outside the cells, and the assignment of these molecules may take a lot of time. Thus, we generally do not recommend this option, and users can choose to employ it based on particular requirements.

## RESULTS

### Datasets and computing resource

We selected two datasets acquired via Stereo-seq technology [4]. One was a mouse liver dataset, a tissue that offers cell boundary information via cell membranes, as in all mammalian tissues. The other dataset was derived from seeds of the plant *Arabidopsis*, a tissue that provides cell boundary information based on rigid cell walls. More details of the two datasets are shown in Table 1.

The experiment for image segmentation was implemented on the STOmics cloud platform [16] with these settings: 32 CPUs, 32 GB memory, and "ALL" resource type. An exception was the watershed method [17], which was implemented using ImageJ on a computer with a 16-core CPU and 16 GB of RAM. Also, the experiment for downstream analysis was implemented on a server with a 40-core CPU, 128 GB of RAM, and 24 GB of GPU.

### Evaluation criteria for cell segmentation performance

In a cell mask image, the gray value of a pixel is set to 255 in the cell area and 0 in the background. True positive (TP, the number of pixels with gray value of 255 in both ground truth and segmented result), true negative (TN, the number of pixels with gray value of 0 in both ground truth and segmented result), false positive (FP, the number of pixels with gray value of 0 in ground truth and 255 in segmented result) and false negative (FN, the number of pixels with gray value of 255 in ground truth and 0 in segmented result) are calculated. The number of cells segmented by a method is $ns$. For each segmented cell (cell$_i$), there should be a corresponding area in the ground truth (area$_i$), where $i$ is the cell index ($i$ = 1, 2, ..., $ns$). The intersection over union metric (IoU) [18] is set as:

$$IoU_i = ao_i/au_i \qquad (1)$$

where ao$_i$ is the overlap area between cell$_i$ and area$_i$, and au$_i$ is the union area of cell$_i$ and area$_i$. Then the precision (Pre), recall (Rec), F1 score (F1_$s$), Dice coefficient (Dc), and average Jaccard index (Avg_$J$) are calculated as:



$$\text{Pre} = \text{TP}/(\text{TP} + \text{FP}) \tag{2}$$

$$\text{Rec} = \text{TP}/(\text{TP} + \text{FN}) \tag{3}$$

$$\text{F1\_}s = 2 \times (\text{Pre} \times \text{Rec})/(\text{Pre} + \text{Rec}) \tag{4}$$

$$\text{Dc} = 2 \times \text{TP}/(2 \times \text{TP} + \text{FP} + \text{FN}) \tag{5}$$

$$\text{Avg\_}J = \sum_{i=1}^{ns} \text{IoU}_i/ns. \tag{6}$$

## Process and evaluation of downstream analysis

The generated single-cell spatial gene expression profiles were input into Stereopy (v0.6.0) [19]. The cells with fewer than ten expressed genes, fewer than three expression counts, and more than 3% mitochondrial genes were removed; genes present in less than three cells were also removed. After normalization, the differentially expressed genes were summarized using Principal Component Analysis to reduce the data dimensionality. Specifically, the number of features was reduced to 10. The Leiden algorithm [20] was used for clustering, and the Uniform Manifold Approximation and Projection (UMAP) algorithm (RRID:SCR_018217) [21] was used to obtain 2D data projections. The Silhouette coefficient (Sc) and Moran's I (MI) were used to evaluate the effect of clustering and the spatial self-correlation of each cluster, respectively. Sc is calculated as:

$$\text{Sc} = (b_j - a_j)/\max\{a_j, b_j\} \tag{7}$$

where $a_j$ is the average distance between the $j$-th sample and other samples in its cluster, and $b_j$ is the average distance between the $j$-th sample and the samples in other clusters. MI is calculated as:

$$\text{MI} = (n/W_0)\left(\sum_{k=1}^{n}\sum_{l=1}^{n}\omega_{k,l}(y_k - \overline{y})(y_l - \overline{y})/\sum_{k=1}^{n}(y_k - \overline{y})^2\right) \tag{8}$$

where $n$ is the number of clusters, $y_k$ and $y_l$ are the attribute values of the $k$-th and $l$-th clusters, respectively, $\overline{y}$ is the mean of all cluster attributes, $\omega_{k,l}$ is the spatial weight between the $k$-th and $l$-th clusters, and $W_0$ is the aggregation of all spatial weights as:

$$W_0 = \sum_{k=1}^{n}\sum_{l=1}^{n}\omega_{k,l}. \tag{9}$$

## STCellbin more accurately segments cells based on cell membrane/wall staining images

We cropped two areas with higher image quality from the two datasets and designed their ground truths based on the manual markup of the cells according to the cell membranes/walls in the staining images. The cell segmentation method of STCellbin was compared with the original Cellpose [18], the state-of-the-art method DeepCell [22], and a traditional watershed method [17].

Using the mouse liver dataset, STCellbin effectively identified cell membranes for segmentation, yielding cell masks that demonstrated acceptable agreement with the stating image and ground truth (Figure 3A, upper). Among all cell mask images, STCellbin provided



the best description of the cell boundaries, outperforming other methods, which missed quite a few cells (Figure 3A, lower). We observed a similar trend using the *Arabidopsis* seed dataset, showing that STCellbin can also effectively identify cell walls for segmentation (Figure 3B). Compared with other methods, STCellbin obtained higher values across most indicators on these two datasets (Figure 3C). The comparison with the original Cellpose validated the effectiveness of STCellbin in adjusting segmentation. While DeepCell is a powerful method, it is primarily designed for segmenting cell nuclei, which involves identifying highlighted areas in the nuclei staining images. This strategy is unsuitable for cell membrane/wall staining images, resulting in less desirable results. Similarly, the traditional watershed method performs poorly on cell membrane/wall staining images. In summary, STCellbin's cell segmentation emerged as the most practical and effective method.

## STCellbin generates more reliable single-cell spatial gene expression profiles for downstream analysis

Currently, there is a lack of image-based one-stop software like STCellbin for Stereo-seq data. Therefore, we compared STCellbin with Baysor (v0.6.2) [23], a tool that generates the spatial gene expression profile without relying on images. However, Baysor could not output results on the complete mouse liver and *Arabidopsis* seed datasets in an acceptable time or a given computational resource. We ran Baysor on a smaller *Arabidopsis* seed dataset, which was the cropped area in the cell segmentation experiment and contained two complete seed data.

The cell area, number of unique genes per cell, and number of gene counts per cell were statistically calculated from the results of STCellbin (Figure 4A). The clustering results of STCellbin were obtained utilizing the generated single-cell spatial gene expression profiles. The clusters of cells were spatially mapped within the tissue (Figure 4B, left-hand side for each tissue), allowing for the observation of their specific positions. From the UMAPs, it was apparent that the different cell types were effectively distinguished (Figure 4B, right-hand side for each tissue). The spatial location of the different cell types positively influenced a series of downstream analyses, such as cellular annotation in less well-studied tissues.

The cells were clustered into seven clusters on the profile of STCellbin, and 14 clusters on the profile of Baysor (Figure 4C, the first subfigure from the left). We observed that the number of cells segmented by Baysor was significantly higher than that segmented by STCellbin, and it did not align with the cell count observed in the ground truth image. This fact could account for the higher number of clusters produced by Baysor. The Sc and MI obtained by both STCellbin and Baysor were not satisfactory (Figure 4C, the second and third subfigures from the left), possibly due to the limited information from a small dataset. Nevertheless, the values from STCellbin were higher than those from Baysor. Moreover, STCellbin demonstrated significant advantages in terms of computing resource usage and running time (Figure 4C, the fourth and fifth subfigures from the left), which also explains why Baysor was unable to process the complete mouse liver and Arabidopsis seed datasets.

It should be noted that the stitching and registration steps of STCellbin could not be performed on the cropped dataset. Hence, the corresponding computational resource usage and running time could not be recorded. Thus, the resource usage and time of STCellbin for comparison were obtained on the complete *Arabidopsis* seed dataset. Specifically, STCellbin was able to process a larger dataset with fewer computational resources and less time



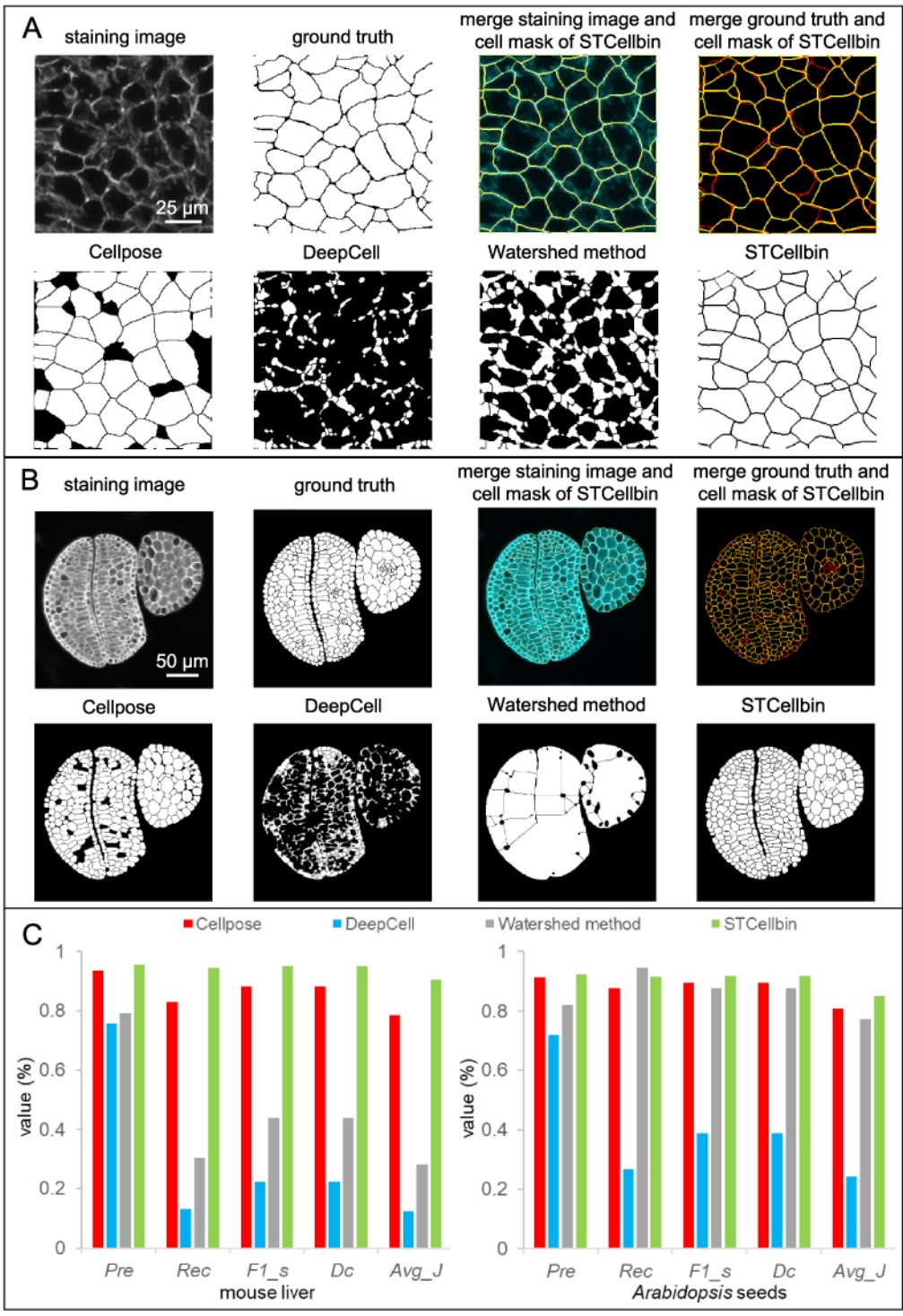

**Figure 3.** Comparison of cell segmentation performance. (A) Cell segmentation results on the cropped area from the mouse liver dataset. In the merged images, cell masks are in yellow, staining images are in cyan, and ground truths are in red. (B) Cell segmentation results on the cropped area from the *Arabidopsis* seed dataset. In the merged images, cell masks are in yellow, staining images are in cyan, and ground truths are in red. (C) Indicator comparison of cell segmentation results on the cropped areas from the mouse liver and *Arabidopsis* seed datasets.

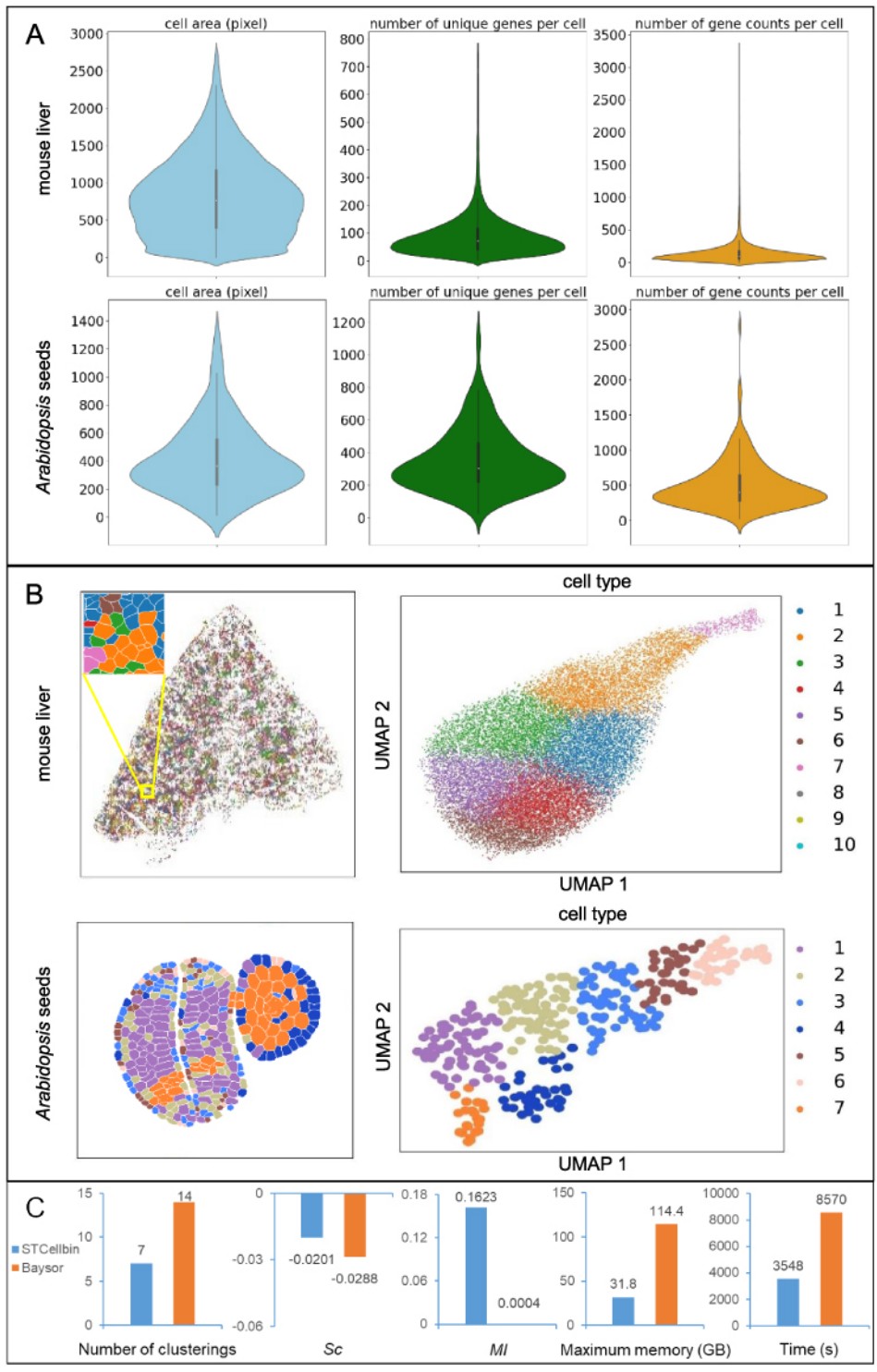

**Figure 4.** Downstream analysis results and comparison. (A) Statistical results of cell area, gene number per cell, and gene expression per cell of STCellbin on the mouse liver dataset and cropped area from the *Arabidopsis* seed dataset. (B) Clustering results and UMAPs of STCellbin on the mouse liver dataset and cropped area from the *Arabidopsis* seed dataset. (C) Indicator comparison of downstream analysis results on the cropped area from the *Arabidopsis* seed dataset, where *Sc* and *MI* are averaged across all clusters.

compared to Baysor. Overall, STCellbin is a more reliable method, particularly for analyzing high-resolution and large-field-of-view spatial transcriptomic data.

## DISCUSSION

Accurate identification of cell boundaries is crucial in generating single-cell resolution in spatial omics applications. Building upon previous work in StereoCell, which uses cell nuclei staining images to generate single-cell spatial gene expression profiles, this STCellbin update extends the capability to automatically process Stereo-seq cell membrane/wall staining images for identifying cell boundaries, thereby facilitating downstream analyses. We also showcased a few examples of the performance of cell membrane/wall segmentation in STCellbin. Currently, the tools for cell nuclei and cell membrane/wall segmentation can be independently executed, allowing users to choose the most suitable solution for their specific applications. In future work, these two techniques could be combined by training a deep learning model compatible with any staining image type, thereby achieving more accurate results.

## AVAILABILITY OF SOURCE CODE AND REQUIREMENTS

- Project name: STCellbin
- Project home page: https://github.com/STOmics/STCellbin
- Operating system(s): Platform independent
- Programming language: Python
- Other requirements: Python 3.8
- License: MIT License
- RRID: SCR_024438

## DATA AVAILABILITY

The data that support the findings of this study have been deposited into the Spatial Transcript Omics DataBase (STOmics DB) of the China National GeneBank DataBase (CNGBdb), with accession number STT0000048. A backup for the data is also provided at the Github link of STCellbin [24]. Archival snapshots of the code are also available from software heritage (Figure 5) [25].

## ABBREVIATIONS

FFT, Fast Fourier Transform; MI, Moran's I; mIF, multiplex immunofluorescence; Sc, Silhouette coefficient; UMAP, Uniform Manifold Approximation and Projection.

## DECLARATIONS

### Ethics approval and consent to participate

The authors declare that ethical approval was not required for this type of research.

### Competing interests

The authors declare that they have no competing interests.

### Authors' contributions

Conceptualization: BZ and ML. Project administration and supervision: SB and XX. Software implementation: ZD, HQ, KS, and HL. Data collection and processing: QK, XF, and LC.

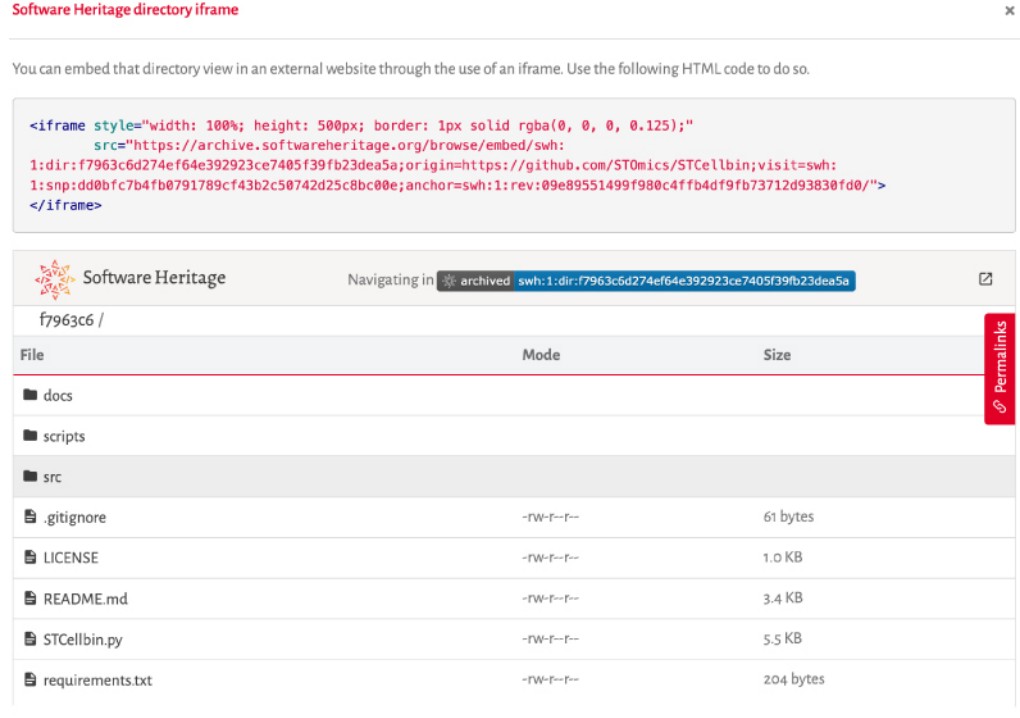

**Figure 5.** The Software Heritage archive of the software [25].
https://archive.softwareheritage.org/browse/embed/swh:1:dir:f7963c6d274ef64e392923ce7405f39fb23dea5a;
origin=https://github.com/STOmics/STCellbin;visit=swh:1:snp:dd0bfc7b4fb0791789cf43b2c50742d25c8bc00e;
anchor=swh:1:rev:09e89551499f980c4ffb4df9fb73712d93830fd0/

Validation: QK and ZD. Project coordination: BZ and ML. Manuscript writing and figure generation: BZ, ML, and QK. Manuscript review: ML, SF, YZ, YL and SB.

## Funding

This work was supported by the National Key R&D Program of China (2022YFC3400400).

## Acknowledgements

We thank China National GeneBank for providing technical support.

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
