## [Editor Report]

Editor’s AssessmentThis paper describes a new spatial transcriptomics method that that utilizes cell nuclei staining images and statistical methods to generate high-confidence single-cell spatial gene expression profiles for Stereo-seq data. STCellbin is an update of StereoCell, now using a more advanced cell segmentation technique, so more accurate cell boundaries can be obtained, allowing more reliable single-cell spatial gene expression profiles to be obtained. After peer review more comparisons were added and more description given on what was upgraded in this version to convince the reviewers. Demonstrating it is a more reliable method, particularly for analyzing high-resolution and large-field-of-view spatial transcriptomic data. And extending the capability to automatically process Stereo-seq cell membrane/wall staining images for identifying cell boundaries.

---

## [Reviewer Report]

Reviewer name and names of any other individual's who aided in reviewerchunquan LiDo you understand and agree to our policy of having open and named reviews, and having your review included with the published manuscript. (If no, please inform the editor that you cannot review this manuscript.)YesIs the language of sufficient quality?YesPlease add additional comments on language quality to clarify if neededIs there a clear statement of need explaining what problems the software is designed to solve and who the target audience is? YesAdditional CommentsIs the source code available, and has an appropriate Open Source Initiative license <a href="https://opensource.org/licenses" target="_blank">(https://opensource.org/licenses)</a> been assigned to the code?YesAdditional CommentsAs Open Source Software are there guidelines on how to contribute, report issues or seek support on the code?YesAdditional CommentsIs the code executable?YesAdditional CommentsIs installation/deployment sufficiently outlined in the paper and documentation, and does it proceed as outlined?Unable to testAdditional CommentsIs the documentation provided clear and user friendly?YesAdditional CommentsAdditional CommentsIs there a clearly-stated list of dependencies, and is the core functionality of the software documented to a satisfactory level?YesAdditional CommentsHave any claims of performance been sufficiently tested and compared to other commonly-used packages? YesAdditional CommentsAdditional CommentsAre there (ideally real world) examples demonstrating use of the software? YesAdditional CommentsAdditional CommentsAny Additional Overall Comments to the AuthorStereo-seq, an advanced spatial transcriptomics technique, allows detailed analysis of large tissues at the single-cell level with precise subcellular resolution. Author's prior software was groundbreaking, generating robust single-cell spatial gene expression profiles using cell nuclei staining images and statistical methods. They've enhanced their software to STCellbin, using cell nuclei images to align cell membrane/wall staining images. This update employs improved cell segmentation, ensuring accurate boundaries and more dependable single-cell spatial gene expression profiles. Successful tests on mouse liver and Arabidopsis seed datasets demonstrate STCellbin's effectiveness, enabling a deeper insight into the role of single-cell characteristics in tissue biology. However, I do have some suggestions and questions about certain parts of the manuscript.   1. The authors should show the advantages and performance of STCellbin compared to other methods, such as its computational efficiency, accuracy, and suitability for various image types.  2. To comprehensively assess the performance of STCellbin, the authors should consider integrating other commonly used cell segmentation evaluation metrics, such as F1-score, Dice coefficient, and so forth. 3. To ensure the completeness and reproducibility of the data analysis, more detailed information regarding the clustering analysis of the single-cell spatial gene expression maps generated through STCellbin is requested. This information should encompass methods, parameters, and results such as cluster type annotations. 4. The authors can use simpler and clearer language and terminology to describe the image registration process in the methods section, ensuring that readers can easily understand the workflow and principles of image registration.
RecommendationMajor Revisions

---

## [Reviewer Report]

Reviewer name and names of any other individual's who aided in reviewerZhaowei WangDo you understand and agree to our policy of having open and named reviews, and having your review included with the published manuscript. (If no, please inform the editor that you cannot review this manuscript.)YesIs the language of sufficient quality?YesPlease add additional comments on language quality to clarify if neededIs there a clear statement of need explaining what problems the software is designed to solve and who the target audience is? YesAdditional CommentsIs the source code available, and has an appropriate Open Source Initiative license <a href="https://opensource.org/licenses" target="_blank">(https://opensource.org/licenses)</a> been assigned to the code?YesAdditional CommentsAs Open Source Software are there guidelines on how to contribute, report issues or seek support on the code?YesAdditional CommentsIs the code executable?YesAdditional CommentsIs installation/deployment sufficiently outlined in the paper and documentation, and does it proceed as outlined?YesAdditional CommentsIs the documentation provided clear and user friendly?YesAdditional CommentsIs there enough clear information in the documentation to install, run and test this tool, including information on where to seek help if required?YesAdditional CommentsIs there a clearly-stated list of dependencies, and is the core functionality of the software documented to a satisfactory level?YesAdditional CommentsHave any claims of performance been sufficiently tested and compared to other commonly-used packages? YesAdditional CommentsAdditional CommentsAre there (ideally real world) examples demonstrating use of the software? YesAdditional CommentsAdditional CommentsAny Additional Overall Comments to the AuthorIn this manuscript, the authors propose STCellbin to generate single-cell gene expression profiles for high-resolution spatial transcriptomics based on cell boundary images. The experiment results on mouse liver and Arabidopsis seed datasets prove the good performance of STCellbin. The topic is significant and the proposed method is feasible. However, there are still some concerns and problems to be improved and clarified.  (1) STCellbin is an update version of StereoCell, but the explanation of StereoCell is not sufficient. The authors should give a more detailed explanation of StereoCell, such as its input and main process.  (2) The authors list some existing dyeing methods in Lines 52-53, Page 3. They should clarify that these methods are used for nuclei staining, which differentiate them from the cell membrane/wall staining methods of following content. It can provide a more accurate explanation for readers and users.  (3) The authors share the GitHub repository of STCellbin, and I noticed that when executing STCellbin, the input only requires the path of image data and spatial gene expression data, the path of the output results, and the chip number. Are there other adjustable parameters?  (4) In Page 5, Line 85, “steps” should be “step”, and in Page 8, Line 145, “must” would be better revised to “should”. Moreover, the writing of “stained image” and “staining image” should be consistent. 
RecommendationMinor Revisions